# Investigation of Genetic Variants Associated with Tryptophan Metabolite Levels via Serotonin and Kynurenine Pathways in Patients with Bipolar Disorder

**DOI:** 10.3390/metabo12111127

**Published:** 2022-11-17

**Authors:** Claudia Pisanu, Alessio Squassina, Pasquale Paribello, Stefano Dall’Acqua, Stefania Sut, Sofia Nasini, Antonella Bertazzo, Donatella Congiu, Anna Meloni, Mario Garzilli, Beatrice Guiso, Federico Suprani, Vittoria Pulcinelli, Maria Novella Iaselli, Ilaria Pinna, Giulia Somaini, Laura Arru, Carolina Corrias, Federica Pinna, Bernardo Carpiniello, Stefano Comai, Mirko Manchia

**Affiliations:** 1Department of Biomedical Science, Section of Neuroscience and Clinical Pharmacology, University of Cagliari, 09042 Cagliari, Italy; 2Unit of Clinical Psychiatry, University Hospital Agency of Cagliari, 09121 Cagliari, Italy; 3Unit of Psychiatry, Department of Medical Sciences and Public Health, University of Cagliari, 09121 Cagliari, Italy; 4Department of Pharmaceutical and Pharmacological Sciences, University of Padua, 35131 Padua, Italy; 5Department of Biomedical Sciences, University of Padova, 35131 Padova, Italy; 6San Raffaele Scientific Institute, 20132 Milano, Italy; 7Department of Psychiatry, McGill University, Montreal, QC H3A 1A1, Canada; 8Department of Pharmacology, Dalhousie University, Halifax, NS B3H 0A2, Canada

**Keywords:** bipolar disorder, GWAS, genome-wide association study, tryptophan, kynurenic acid, kynurenine, quinolinic acid, SNP

## Abstract

The kynurenine pathway (KP) may play a role in the pathophysiology of bipolar disorder (BD). We conducted a genome-wide association study (GWAS) to identify genetic variants associated with the plasma levels of the metabolites of tryptophan (TRP) via the serotonin (5-HT) and kynurenine (KYN) pathways in 44 patients with BD and 45 healthy controls. We assessed whether variants that were differentially associated with metabolite levels based on the diagnostic status improved the prediction accuracy of BD using penalized regression approaches. We identified several genetic variants that were significantly associated with metabolites (5-HT, 5-hydroxytryptophan (5-HTP), TRP, and quinolinic acid (QA) or metabolite ratios (5-HTP/TRP and KYN/TRP) and for which the diagnostic status exerted a significant effect. The inclusion of genetic variants led to increased accuracy in the prediction of the BD diagnostic status. Specifically, we obtained an accuracy of 0.77 using Least Absolute Shrinkage and Selection Operator (LASSO) regression. The predictors retained as informative in this model included body mass index (BMI), the levels of TRP, QA, and 5-HT, the 5-HTP/TRP ratio, and genetic variants associated with the levels of QA (rs6827515, rs715692, rs425094, rs4645874, and rs77048355) and TRP (rs292212) or the 5-HTP/TRP ratio (rs7902231). In conclusion, our study identified statistically significant associations between metabolites of TRP via the 5-HT and KYN pathways and genetic variants at the genome-wide level. The discriminative performance of penalized regression models incorporating clinical, genetic, and metabolic predictors warrants a follow-up analysis of this panel of determinants.

## 1. Introduction

Bipolar disorder (BD) is a chronic psychiatric disorder characterized by recurring episodes of depression and hypomania/mania, alternating with intervals of well-being (euthymic phase), and affects approximately 1–2% of individuals worldwide. Although mania/hypomania is the most recognizable characteristic of BD and frequently the first episode of polarity in BD, depression is the most frequent clinical presentation. Of note, there is consistent evidence that a poor response to antidepressants prescribed in acute depressive phases might be a predictor of later bipolarity [1,2]. The treatment of BD with mood stabilizers and/or antipsychotics has been successful in reducing recurrence/relapse rates, but a significant proportion of BD patients have poor outcomes, and safety concerns might limit their acceptability. Despite progress in the comprehension of the pathophysiology of BD, there has been little advancement in the development of novel effective and safer treatments. The kynurenine pathway (KP) represents an interesting biochemical pathway that may play a role in BD pathophysiology and treatment response [3,4,5,6,7,8]. The essential amino acid tryptophan (TRP) is mostly metabolized via the serotonin (5-HT) and kynurenine (KYN) pathways [4]. Because of the activity of the enzymes tryptophan 2,3-dioxygenase (TDO) and indoleamine 2,3-dioxygenase (IDO), TRP is converted into KYN, the central metabolite of the metabolic route (Figure 1). KYN is then metabolized into kynurenic acid (KYNA) by the enzyme kynurenine aminotransferase (KAT) or into 3-hydroxykynurenine (3-HK) by the enzyme kynurenine 3-monooxygense (KMO). 3-HK, via the action of kynureninase (KYNU), is converted into 3-hydroxyanthranilic acid, which, via the action of the 3-hydroxyanthranilic acid oxygenase enzyme, is transformed into 2-amino-3-carboxymuconate semialdehyde, which undergoes non-enzymatic cyclization to quinolinic acid (Figure 1). Many other different metabolites are also formed along the KP, but these are those associated with BD pathophysiology and treatment. In mood disorders, including BD, stress, the activation of the hypothalamic–pituitary–adrenal axis, and inflammation stimulate either or both TDO and IDO enzymes [4,9,10], with a subsequent increase in the amount of TRP degraded via the KP at the expense of the brain availability of TRP for the synthesis of 5-HT. In addition, the activation of the KP leads to an imbalance in the formation of downstream metabolites of KYN, such as neurotoxic QUIN or 3-HK and neuroprotective KYNA. Overall, although there is not yet full agreement among different studies, this hypothesis may reconcile, on the one hand, the 5-HT deficiency and, on the other hand, the neuroinflammatory/neurotoxicity hypotheses of mood disorders [4,10,11]. Interestingly, the pathophysiological underpinnings of treatment-resistant forms of depression (often predicting later BD) appear to include alterations in the tryptophan and kynurenine pathway [12].

Changes in the different metabolites of the 5-HT and KYN pathways, which may increase the likelihood of developing BD, could also depend on malfunctions of the enzymes involved in the two metabolic pathways, which may be of genetic origin. BD is highly heritable and polygenic, and some studies have already demonstrated a possible association between single-nucleotide polymorphisms (SNPs) in the genes involved in the 5-HT and KYN pathways and BD. For instance, a functional genetic variant of KMO associated with KYNA levels in the cerebrospinal fluid increased the risk of the manifestation of psychotic symptoms during mania in BD patients [13]. The frequency of the haplotype of the KAT III gene CGCTCT, which may affect the levels of KYNA, is significantly higher in BD but also in major depressive disorder (MDD) patients than in the control population [14]. Different studies also showed a possible association between polymorphisms in the *TPH2* gene and BD. In particular, in the meta-analysis by Ottenhof et al. [15], it was found that the A allele of SNP rs11178997 and haplotypes containing this allele of the *TPH2* gene were associated with the diagnosis of BD. However, in these studies, it was not investigated whether the association between a polymorphism in one of the genes encoding for enzymes of the 5-HT or KYN pathway and BD was mediated by an effect given by the circulating levels of the metabolites formed by that specific enzyme. 

In this context, here, we aimed to assess the possible association between circulating levels of the metabolites of TRP via the 5-HT and KYN pathways and the genetic architecture of the enzymes involved in the different steps of these two metabolically connected pathways. Ultimately, we expected to gain a better understanding of the contribution of these pathways to the pathophysiology of BD, which is central in the development of novel therapies, as well as in the identification of biomarkers of disease. 

## 2. Materials and Methods

### 2.1. Sample

This study is part of an ongoing project aimed at exploring the longitudinal relationship between the melatonin system and BD in a cohort of 50 patients with BD and 50 healthy controls (HCs), as previously described [6,16]. Participants were recruited consecutively from 2019 to 2021 at the Psychiatric Unit of the University Hospital of Cagliari and of the Department of Medical Sciences and Public Health, University of Cagliari. In the present study, we used only cross-sectional data from 46 patients with BD in the euthymic state and 47 HCs, for which genome-wide genotyping data were available. Two patients and 2 HCs were excluded after quality control of genetic data (see following sections), leading to a final sample of 44 patients of BD and 45 HCs. The diagnosis of BD was made by trained psychiatrists in accordance with the Diagnostic and Statistical Manual of Mental Disorders (DSM)-5 criteria [17] and applying the Italian version of the Structured Clinical Interview (SCID). Inclusion and exclusion criteria for patients with BD and HCs have been previously described [6]. Briefly, patients aged between 18 and 65 years with a diagnosis of BD type 1 or type 2 were included, while exclusion criteria comprised a current and/or lifetime diagnosis of other psychiatric or neurological disorders, substance-use disorders, or other severe unregulated medical conditions, a history of traumatic brain insults, or being treated with melatonergic compounds (melatonin and/or agomelatine) for at least two months before enrollment. 

Sex- and age-matched controls with no history of psychiatric disorders, neurological disorders, or other severe unregulated medical conditions were also recruited. Patients and controls were all Caucasians and of Italian origin. For each participant, fasting blood was collected in the morning in EDTA-containing tubes. The research protocol followed the principles of the Declaration of Helsinki and was approved by the Ethics Committee of the University Hospital Agency of Cagliari (approval number: PG/2019/6277). All participants provided written informed consent after a detailed description of the study procedures.

### 2.2. Measurement of Plasma Levels of TRP and Its Metabolites via 5-HT and KYN

Blood samples were centrifuged at 2500 rpm and 4 °C for 10 min. Plasma was aliquoted and stored frozen at −80 °C until the analysis. Plasma levels of TRP and its metabolites via the 5-HT and KYN pathways were determined within 4 months from collection, according to standard methods in our lab [5,6,18]. TRP, 5-HTP, 5-HT, and KYN levels were determined using an HPLC system equipped with UV–Vis and fluorometric detectors, whereas QA, KYNA, 3-HK, and MLT were determined by LC-MS/MS by using alfa-methyltryptophan as an internal standard. Details on the methodology have been previously published [5,6,18].

### 2.3. Genotyping and Quality Control 

Genome-wide genotyping was carried out at the Centro Servizi di Ateneo per la Ricerca (CeSAR) of the University of Cagliari using Global Screening Arrays (Illumina) according to the manufacturer’s protocols. Quality control procedures were conducted using PLINK v. 1.9 [19]. Autosomal SNPs with low levels of missingness (>80%), without showing deviations from Hardy–Weinberg Equilibrium (*p* > 10^−6^), and with a minor allele frequency (MAF) higher than 5% were retained. Individuals with low genotype rates (<0.8), sex inconsistencies, a heterozygosity rate deviating more than 3 sd from the mean, or cryptic relatedness (pihat > 0.2) were excluded. Population stratification was checked with multidimensional scaling in PLINK. Data were imputed on the Michigan server [20] with Minimac 4, using 1000 G Phase 3 v5 (European population) as the reference panel. After imputation, SNPs with MAF < 5% or low imputation quality (R-square < 0.3) were excluded, and analyses were conducted on 5,809,187 SNPs.

### 2.4. Statistical Analysis

The normality of variables was assessed using the Shapiro–Wilk test. Differences in metabolite levels or ratios between patients with BD and HCs were computed using analysis of covariance (ANCOVA), adjusting for age, gender, and body mass index (BMI). 

The association between SNPs and metabolite levels was analyzed with linear regression models implemented in PLINK v. 1.9, adjusting for age, gender, BMI, and diagnostic status (patient with BD or control). Results with *p*-values < 4.99 × 10^−8^ were considered to be significant. The annotation of identified variants, the identification of lead SNPs, and the evaluation of their functional effects were conducted using the FUMA platform [21]. For SNPs for which the effect of diagnosis was significant (at a *p* < 0.05), we repeated analyses separately in patients with BD and controls. In addition to genome-wide analyses, we report results for specific candidate genes encoding enzymes known to be involved in the determination of metabolite levels or metabolite ratios: the indoleamine 2,3-dioxygenase (IDO1), indoleamine 2,3-dioxygenase 2 (IDO2), and tryptophan 2,3-dioxygenase (TDO2) genes for the kynurenine/tryptophan ratio; the kynurenine 3-monooxygenase (KMO) gene for the 3-hydroxykynurenine/kynurenine ratio; the kynurenine–oxoglutarate transaminase (KYAT2) gene for the kynureninc acid/kynurenine ratio; and the kynureninase (KYNU) gene for levels of the 3-hydroxykynurenine and quinolinic acid metabolites.

To assess whether the inclusion of genetic variants associated with metabolite levels can improve the prediction accuracy of BD compared with a model only including levels of metabolites or metabolite ratios, we used two penalized regression approaches: Least Absolute Shrinkage and Selection Operator (LASSO) and ridge regression. Penalized regression approaches help to prevent overfitting and multicollinearity issues by imposing a constraint on the loss function. LASSO regression allows the selection of parsimonious models by shrinking the estimates of some coefficients to 0 [22]. Variables with regression coefficients equal to 0 after shrinkage are excluded from the model. Ridge regression shrinks the estimates to nearly 0 and retains all variables in the model. We randomly divided our dataset into training and test partitions with a 60:40 ratio. The λ parameter was chosen with 10-fold cross-validation. For each method, we computed two models of increasing complexity, both with the diagnosis of BD as the outcome. Model 1 included age, gender, BMI, and levels of metabolites or metabolite ratios that we identified to be significantly different in patients with BD compared with controls in the ANCOVA analysis previously described. Model 2 included the predictors of Model 1 as well as genetic variants that, in the analyses conducted with PLINK, were significantly associated with levels of the metabolites included in Model 1 and for which we detected a significant contribution of diagnosis in the model. We then compared the accuracy of the two models to assess whether the inclusion of genetic variants might lead to a better prediction of the BD status compared with the model including only metabolites of the examined pathways. LASSO and ridge regression analyses were performed with the glmnet and caret packages in R [23]. For both models, we computed the area under the curve (AUC) and plotted Receiver Operating Characteristic (ROC) curves using the pROC R package. 

## 3. Results

### 3.1. Association Analysis of Metabolite Levels with Genetic Variants

The demographic and clinical characteristics of the sample are summarized in Table 1. In keeping with our previous study, patients with BD showed lower TRP, lower QA, and higher 5-HTP and 5-HT levels compared with HCs [6]. In addition, patients with BD showed a higher 5-HTP/TRP ratio compared with HCs (Table 1). 

SNPs that were significantly associated with metabolites or metabolite ratios at a genome-wide significant threshold and for which being a patient or control, and thus diagnosis, exerted a significant effect are reported in Table 2. Analyses conducted separately in patients and controls generally showed stronger associations between these SNPs and metabolites or metabolite ratios in controls compared with patients. Of note, several SNPs were significantly associated with levels of QA exclusively in controls (Table 2). 

SNPs significantly associated with metabolites or metabolite ratios at a genome-wide significant threshold for which being a patient or control exerted no significant effect are reported in Table 3.

No SNPs were significantly associated with the level of KYN or the 3-HK/KYN ratio. In addition, analyses focused on specific candidate genes did not show any significant association between SNPs located in genes encoding the IDO1, IDO2, TDO2, KMO, KYAT2, and KYNU enzymes and metabolite levels or ratios after multiple-testing correction. SNPs showing nominally significant associations are reported in Appendix A. A significant effect of the diagnosis was detected in the association between SNPs located in the KYNU gene and QA levels. However, stratified analyses did not show nominal associations between these SNPs and either patients or controls.

### 3.2. Penalized Regression with Diagnosis of BD as the Outcome

Finally, we used penalized LASSO and ridge regression to identify whether the addition of genetic markers to metabolite levels might lead to increased accuracy in the prediction of the diagnosis of BD. To this aim, two models were created: Model 1 included metabolites showing significantly different levels in patients with BD compared with controls (Table 1) adjusted for age, gender, and BMI; Model 2 also included genetic variants that were significantly associated with the levels of the metabolites included in Model 1 and for which we detected a significant contribution of the diagnosis in the model (Table 2). In Model 1, we obtained an accuracy of 0.71 with either ridge or LASSO regression. Predictors retained by LASSO regression as informative included BMI and the levels of the three metabolites TRP, QA, and 5-HT, as well as the 5-HTP/TRP ratio. 

In Model 2, we obtained an accuracy of 0.74 and 0.77 with ridge and LASSO regression, respectively. Predictors retained by LASSO regression as informative included BMI, the levels of the three metabolites TRP, QA, and 5-HT, the 5-HTP/TRP ratio, and seven genetic variants for which we detected a significant effect of the diagnostic status on their association with levels of QA (rs6827515, rs715692, rs425094, rs4645874, and rs77048355), TRP (rs292212), or the 5-HTP/TRP ratio (rs7902231). 

The results of the two penalized regression approaches suggest that the inclusion of genetic variants differentially affecting metabolite levels in patients compared with controls can increase the accuracy of the prediction of the BD diagnostic status. However, we found a similar AUC for ridge regression for the two models (Figure 2A,B) and a higher AUC for the LASSO regression model including only metabolite levels (Figure 2C) compared to the one also including genetic predictors (Figure 2D).

## 4. Discussion

In this study, we identified genetic variants differentially affecting the circulating levels of the metabolites of TRP via the 5-HT and KYN pathways in patients with BD and non-psychiatric controls. Indeed, we observed several genetic variants associated with metabolite levels at a genome-wide threshold for which the diagnostic status played a significant role. Specifically, we found several variants associated with increased levels of QA in HCs but not in patients affected by BD (Table 2). Using penalized regression models, a model including the levels of QA together with TRP, 5-HT, and the 5-HTP/TRP ratio was able to discriminate patients with BD from HCs with good accuracy. Moreover, this accuracy increased when the identified genetic variants were included in the model. The finding that patients with BD showed lower levels of QA compared with controls is in contrast to the commonly believed hypothesis of an imbalance towards the formation of downstream neurotoxic metabolites of KYN in mood disorders. However, it is worth mentioning that a recent meta-analysis did not find significant differences in QA levels between patients with BD and controls [3]. Among the five studies included in the meta-analysis, three did not find significant differences in QA levels, while two observed lower QA levels in patients with BD or MDD compared with controls [24,25], in line with our results. While a potential explanation might be related to the fact that QA levels could be decreased due to its further degradation to NAD [24], it is also possible that clinical or genetic factors might contribute to the variability in QA levels observed across studies. Indeed, QA was the metabolite for which we identified the highest number of associated genetic variants, as well as of variants for which the association with QA levels was dependent on the diagnostic status. Finally, it is yet to be clarified whether QA levels may vary according to the phase of the disorder. Another set of SNPs located in the *PLEKHG4B*, *CAMK1D*, and *TMEM266* genes was found to be associated with the level of KYNA. Of these, the finding on *CAM1KD* appears to be of great interest. This gene encodes for a calcium-/calmodulin-dependent protein kinase (subtype 1D) and was found to be associated with the enzyme activity of Ndel1, a DISC1-interacting oligopeptidase associated with both neuronal migration and neurite outgrowth, in a sample of 83 patients with schizophrenia and 92 HCs [26]. Interestingly, the association between CAMK1D and Ndel1 enzyme activity was independent of the clinical status [26], as in our study. The same gene was found to be associated with the manifestation of vomiting during lithium treatment in patients with BD, making it a promising candidate for safety pharmacogenomics [27]. Finally, the *TIAM1* gene (TIAM Rac1 Associated GEF 1), encoding for a RAC1-specific guanine nucleotide exchange factor, was associated with the QA/KYNA ratio in our sample of patients with BD and HCs. This gene is critical for neuronal morphogenesis and neurite outgrowth and has been found to be associated with the presentation of developmental delay, intellectual disability, speech delay, and seizures [28].

Another aim of our study was to evaluate the association between circulating levels of the metabolites of the TRP via the 5-HT and KYN pathways and common variants of genes encoding the enzymes involved in the different steps of these two pathways. After multiple-testing correction, we did not identify any statistically significant association between variants located in these genes and metabolite levels. Our results do not support a relevant role for common genetic variants located in these genes in driving differences in the levels of the metabolites of these two pathways. 

Among the genetic variants found to be more informative in the prediction of the diagnosis of BD in LASSO regression models, rs425094 resides in the *NRG1* gene. This gene encodes a membrane glycoprotein that plays a crucial role in cell–cell signaling and in the development of multiple organ systems. In addition to being previously associated with BD [29], depression [30], and schizophrenia [31], the expression levels of the *NRG1* gene in different brain regions were associated with matter abnormalities, clinical symptoms, and cognition in a recent study conducted in a transdiagnostic psychiatric cohort [31]. Other variants were located in intergenic regions or in genes that have not been previously associated with psychiatric disorders (e.g., *COMMD10*) or for which the function is not yet known (*RP11-25I11.1*).

Our findings should be interpreted in the context of several limitations. First, our sample size was limited, and we might not have been able to detect statistically significant associations of moderate/small effect size between genes and metabolite levels. However, some of our findings present very high significance levels, pointing to their robustness and possible replicability in additional samples. Secondly, we performed a targeted analysis of metabolite levels instead of a hypothesis-free untargeted approach. This might have limited our ability to detect associations with other components that are directly or indirectly related to the TRP pathway. However, our a priori hypothesis was based on the investigation of TRP via the 5-HT/MLT and KYN pathways, which are thought to play a substantial role in the pathogenesis of BD [16]. Third, the cross-sectional design might have impacted the robustness of clinical data. It should be noted, however, that these patients are part of a naturalistic cohort followed up at our center in certain cases for more than a decade. Finally, the limited sample size might have impacted the performance of the predictive models. However, the application of ridge and LASSO penalized regressions was able to counterbalance, to some extent, the lack of adequate statistical power.

In conclusion, our study has identified statistically significant associations between metabolites of TRP via 5-HT and KYN pathways and genetic variants at the genome-wide level. The discriminative performance of the penalized regression models incorporating multi-layer predictors (clinical, genetic, and metabolic) warrants a follow-up analysis of this panel of determinants in future independent studies.

## Figures and Tables

**Figure 1 metabolites-12-01127-f001:**
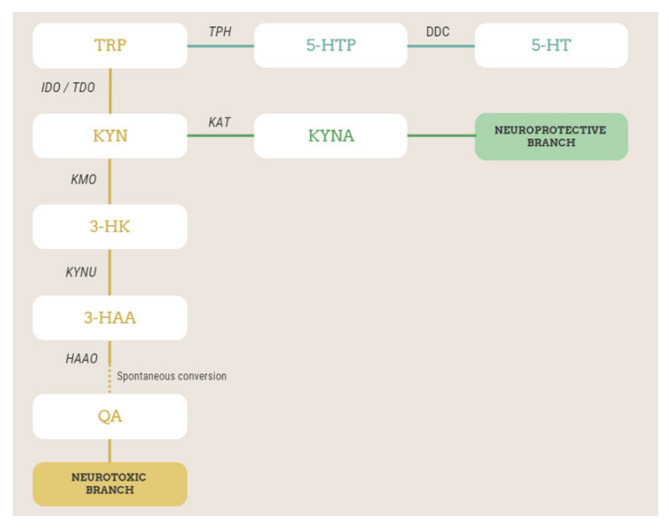
Simplified scheme of TRP metabolism via kynurenine and serotonin pathways, with the metabolites and enzymes studied in this work highlighted. Metabolites in the orange branch are neurotoxic, while those in the green branch are neuroprotective. TRP, tryptophan; 5-HTP, 5-hydroxytryptophan; TPH, tryptophan hydroxylase; DDC, L-amino-acid decarboxylase; 5-HT, serotonin or 5-hydroxytryptamine; KYN, kynurenine; IDO, indoleamine 2,3-dioxygenase; TDO, tryptophan 2,3-dioxygenase; KATs, kynurenine aminotransferases; KYNA, kynurenic acid; KMO, kynurenine 3-monooxygenase; 3-HK, 3-hydroxykynurenine; KYNU, kynureninase; 3-HAA, 3-hydroxyanthranilic acid; 3-HAAO, 3-hydroxyanthranilate 3,4-dioxygenase; QA, quinolinic acid.

**Figure 2 metabolites-12-01127-f002:**
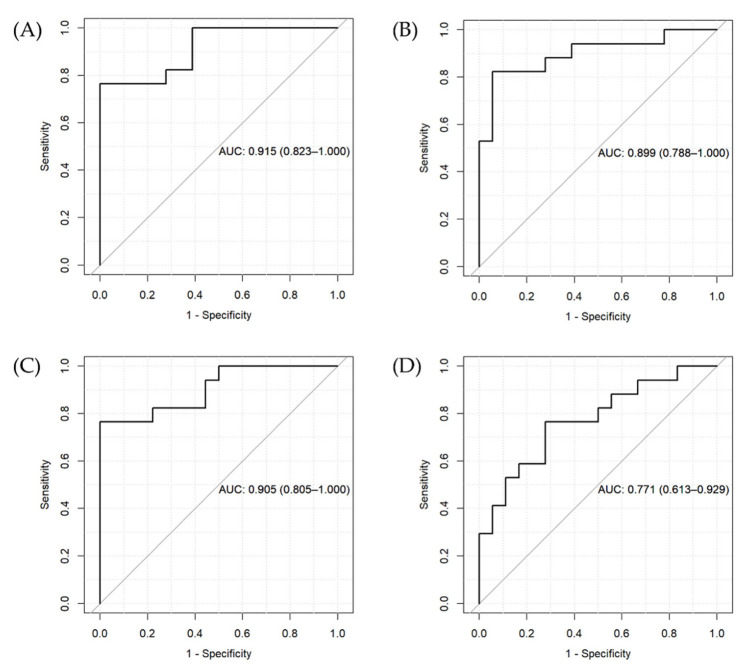
ROC curves and AUC for the penalized regression models. (**A**) Ridge regression for Model 1 (including metabolite levels, age, gender, and BMI), (**B**) ridge regression for Model 2 (including variables in Model 1 + genetic variants), (**C**) LASSO regression for Model 1, and (**D**) LASSO regression for Model 2.

**Table 1 metabolites-12-01127-t001:** Characteristics of the sample.

	Patients with BD (n = 44)	HCs (n = 45)	Statistics	*p*
Age (years)	51.94 (±11.01)	50.99 (±8.06)	t = −0.47	0.64
Sex (% F)	63.6	64.4	X^2^ = 0.00	1.00
BMI	25.77 (±5.23)	23.66 (±3.52)	**t = 2.24**	**0.028**
TRP [μg/mL]	9.41 (±1.64)	11.15 (±1.7)	**F = 24.74**	**3.4 × 10^−6^**
KYN [μg/mL]	0.35 (±0.10)	0.39 (±0.14)	F = 2.87	0.09
KYNA [ng/mL]	9.28 (±4.66)	10.53 (±8.27)	F = 0.77	0.38
3-HK [ng/mL]	42.55 (±2.94)	44.16 (±13.36)	F = 0.66	0.42
QA [ng/mL]	143.49 (±15.09)	159.37 (±46.51)	**F = 4.65**	**0.034**
5-HTP [ng/mL]	87.36 (±30.38)	69.78 (±35.26)	**F = 6.33**	**0.014**
5-HT [ng/mL]	324.02 (±175.24)	245.18 (±147.00)	**F = 5.23**	**0.025**
MLT [pg/mL]	10.97 (±5.90)	11.95 (±5.90)	F = 0.61	0.44
KYN/TRP × 1000 ratio	38.77 (±14.93)	35.75 (±11.76)	F = 1.10	0.30
3-HK/KYN ratio	131.82 (±39.93)	128.1 (±66.29)	F = 0.10	0.75
KYNA/KYN ratio	28.8 (±16.72)	29.48 (±25.85)	F = 0.02	0.89
QA/KYNA ratio	22.57 (±19.00)	22.34 (±14.44)	F = 0.00	0.95
5-HTP/TRP ratio	9.44 (±3.35)	6.34 (±3.25)	**F = 19.86**	**2.6 × 10^−5^**

For metabolites and metabolite ratios, results are from ANCOVA analyses with metabolite levels as outcome and diagnosis as predictor, adjusting for age, gender, and BMI. Abbreviations: 3-HK, 3-hydroxykynurenine; 5-HT, serotonin; 5-HTP, 5-hydroxytryptophan; BD, bipolar disorder; HCs, healthy controls; KYN, kynurenine; KYNA, kynurenic acid; MLT, melatonin; QA, quinolinic acid; TRP, tryptophan. Significant results are reported in bold. Data for age and metabolite levels or ratios are reported as means (±standard deviations).

**Table 2 metabolites-12-01127-t002:** SNPs significantly associated with metabolite levels, with a significant effect of diagnosis.

					All		BD Patients			Controls		
CHR	SNP	Gene	EA	OA	BETA	*p*	EAF	BETA	*p*	EAF	BETA	*p*
** *5-HT* **												
3	rs73153916	*RP11-23D24.2*	T	C	275.6	1.2 × 10^−9^	0.08	259.8	6.1 × 10^−6^	0.03	322.9	1.0 × 10^−4^
** *5-HTP* **												
3	rs11130047	Intergenic	A	G	62.96	2.5 × 10^−8^	0.05	32.26	0.08	0.06	84.04	8.4 × 10^−9^
10	rs7902231	Intergenic	A	C	58.09	1.9 × 10^−9^	0.06	39.15	0.008	0.08	73.25	2.5 × 10^−8^
** *TRP* **												
5	rs292212	Intergenic	A	G	2.70	4.1 × 10^−8^	0.07	2.37	0.001	0.06	3.06	2.5 × 10^−5^
** *5-HTP/TRP* **												
2	rs10209883	Intergenic	A	G	3.07	4.0 × 10^−8^	0.26	3.02	6.8 × 10^−5^	0.21	3.28	2.0 × 10^−4^
10	rs7902231	Intergenic	A	C	6.11	1.2 × 10^−10^	0.06	5.74	3.0 × 10^−4^	0.08	6.48	7.7 × 10^−8^
** *KYN/TRP* **												
3	rs13063065	*ZNF385D*	G	T	19.61	3.6 × 10^−8^	0.07	26.3	1.9 × 10^−4^	0.13	17.17	3.3 × 10^−5^
** *QA* **												
4	rs6827515	Intergenic	A	G	51.62	3.6 × 10^−8^	0.08	8.92	0.16	0.06	90.35	1.7 × 10^−8^
5	rs715692	*COMMD10*	C	G	59.27	6.5 × 10^−9^	0.06	11.39	0.12	0.04	98.90	4.0 × 10^−9^
7	rs116926743	Intergenic	C	G	57.90	2.2 × 10^−8^	0.06	5.03	0.51	0.04	96.33	1.3 × 10^−8^
8	rs425094	*NRG1*	C	T	47.19	3.0 × 10^−8^	0.08	3.89	0.54	0.09	76.56	3.7 × 10^−8^
12	rs10843992	*ETFBKMT*	C	T	59.01	6.3 × 10^−9^	0.05	-5.73	0.50	0.06	95.82	1.7 × 10^−9^
14	rs4645874	Intergenic	T	C	38.38	3.2 × 10^−8^	0.15	7.88	0.17	0.11	58.13	1.9 × 10^−7^
17	rs117632415	Intergenic	A	G	64.48	3.9 × 10^−10^	0.05	3.05	0.73	0.06	97.57	1.4 × 10^−9^
18	rs77048355	*RP11-25I11.1*	A	G	59.06	1.0 × 10^−8^	0.08	3.60	0.63	0.03	119.6	3.1 × 10^−13^

Abbreviations: 5-HT, serotonin; 5-HTP, 5-hydroxytryptophan; CHR, chromosome; EA, effect allele; EAF, effect allele frequency; KYN, kynurenine; OA, other allele; QA, quinolinic acid; SNP, single-nucleotide polymorphism; TRP, tryptophan.

**Table 3 metabolites-12-01127-t003:** SNPs associated with metabolite levels with no significant effect of diagnosis.

CHR	SNP	Gene	EA	OA	EAF	BETA	*p*
** *5-HT* **							
1	rs10913387	Intergenic	T	G	0.23	152.3	4.2 × 10^−8^
** *3-HK* **							
13	rs9579535	Intergenic	A	G	0.06	15.27	3.2 × 10^−8^
** *KYNA* **							
5	rs115379324	*PLEKHG4B*	A	G	0.05	11.55	4.6 × 10^−8^
10	rs74119224	*CAMK1D*	A	G	0.06	11.20	1.3 × 10^−8^
12	rs112998010	Intergenic	T	C	0.06	10.61	4.2 × 10^−9^
15	rs62030183	*C15orf27 (TMEM266)*	T	C	0.06	11.34	2.9 × 10^−10^
** *QA* **							
1	rs17466728	*RP5-884C9.2*	A	T	0.06	57.01	5.8 × 10^−9^
3	rs80184502	*CD96*	C	A	0.05	60.55	4.2 × 10^−9^
7	rs138032435	Intergenic	A	G	0.05	57.51	1.9 × 10^−8^
9	rs111735019	Intergenic	T	A	0.06	55.53	1.5 × 10^−8^
12	rs200629811	Intergenic	T	C	0.05	56.66	3.9 × 10^−8^
13	rs9556820	Intergenic	T	C	0.08	49.65	4.0 × 10^−10^
16	rs77023911	Intergenic	T	C	0.06	53.21	3.3 × 10^−8^
16	rs62041963	*RP11-467I17.1*	T	C	0.10	47.39	6.8 × 10^−10^
17	rs72842909	*ALOX15B*	T	C	0.07	51.96	2.8 × 10^−8^
17	rs184751267	Intergenic	T	C	0.05	63.07	6.6 × 10^−10^
19	rs73041282	*YIF1B*	A	G	0.06	59.95	1.2 × 10^−9^
19	rs113703374	*IFNL3*	A	G	0.06	57.49	5.1 × 10^−9^
** *KYNA/KYN* **							
1	rs1888826	*ADCK3*	T	C	0.10	24.77	4.0 × 10^−8^
2	rs141988608	Intergenic	T	C	0.07	32.59	1.4 × 10^−8^
9	rs10810150	Intergenic	T	C	0.08	30.17	1.8 × 10^−8^
10	rs12413445	Intergenic	C	G	0.06	33.67	4.7 × 10^−8^
13	rs9580486	Intergenic	T	C	0.05	36.47	1.1 × 10^−8^
14	rs2031065	Intergenic	C	T	0.05	39.93	3.6 × 10^−8^
15	rs32030170	*C15orf27 (TMEM266)*	T	C	0.06	34.54	7.7 × 10^−9^
** *QA/KYNA* **							
12	rs73041300	LINC00942	A	G	0.07	25.46	1.3 × 10^−8^
13	rs9587565	Intergenic	A	G	0.06	28.11	3.4 × 10^−9^
15	rs1967975516	Intergenic	G	C	0.06	29.56	1.1 × 10^−10^
21	rs138622229	*TIAM1*	A	C	0.05	35.14	2.1 × 10^−13^
** *MLT* **							
5	rs6862509	*AC011343.1*	C	G	0.06	9.89	2.7 × 10^−8^

Abbreviations: 3-HK, 3-hydroxykynurenine; 5-HT, serotonin; CHR, chromosome; EA, effect allele; EAF, effect allele frequency; KYN, kynurenine; KYNA, kynurenic acid; OA, other allele; QA, quinolinic acid; SNP, single-nucleotide polymorphism.

## Data Availability

The data presented in this study are available upon request from the corresponding author because of ongoing analysis.

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
