# Peer review of "Investigation of Genetic Variants Associated with Tryptophan Metabolite Levels via Serotonin and Kynurenine Pathways in Patients with Bipolar Disorder"

_metabolites, 2022, doi:10.3390/metabo12111127_

Round 1
Reviewer 1 Report
Review of the manuscript:
Investigation of genetic variants associated with levels of metabolites of the tryptophan to the serotonin and kynurenine pathways in patients with bipolar disorder
Authors: Claudia Pisanu, Alessio Squassina, Pasquale Paribello , Stefano Dall’Acqua , Stefania Sut, Sofia Nasini, Antonella Bertazzo, Donatella Congiu , Anna Meloni, Mario Garzilli, Beatrice Guiso, Federico Supran, Vittoria Pulcinelli, Maria Novella Iaselli, Ilaria Pinna, Giulia Somaini, Laura Arru, Carolina Corrias, Federica Pinna, Bernardo Carpiniello, Stefano Comai, Mirko Manchia
Journal: Metabolites mdpi
The article is written very well. The authors wrote a very good research article of high quality. It is obvious that the authors are very well versed in the issue, with ease and pedagogical mastery they provide not only a literary overview, discussion, but also results that are statistically very well evaluated. The authors obtained interesting and original results; e.g. they identified genetic variants affecting circulating levels of the metabolites of the TRP to 5-HT and KYN pathways in a differential way in patients with BD and non-psychiatric controls, and observed several genetic variants associated with metabolite levels at a genome-wide threshold for which the diagnostic status exerted a significant role. They also evaluated the association between circulating levels of the metabolites of the TRP to 5-HT and KYN pathways and common variants of genes encoding the enzymes involved in the different steps of these two pathways.
Conclusion:
The work is of high quality, the results are original, I have no comments.
The article is suitable for publication.

Author Response
We are very grateful to the reviewer for the positive assessment of our work.
Reviewer 2 Report
The manuscript is well written, and presented very interesting information, except for minor issues.
· Keywords: shouldn’t include any abbreviations such as GWAS, SNP.
· Check Plagiarism as it is high 37%
Author Response
Q1) Keywords: shouldn’t include any abbreviations such as GWAS, SNP.
R1) We removed the keyword SNP, while we believe that the keyword GWAS should be included among keywords, as it is a common term that researchers use to define this type of study and therefore is widely searched and needed to retrieve the article.
Q2) Check Plagiarism as it is high 37%
R2) We checked the text of the manuscript for plagiarism with the Grammarly checker and did not identify the percentage reported by the Reviewer. We found plagiarism to be 5% and related to either definitions or methodological sentences that can be related to different studies (e.g. “Participants were recruited consecutively at the Psychiatric Unit of the University Hospital of Cagliari and of the Department of Medical Sciences and Public Health”).
Reviewer 3 Report
The manuscript entitled "Investigation of genetic variants associated with levels of metabolites of the tryptophan to the serotonin and kynurenine pathways in patients with bipolar disorder” was well prepared. Both the "Introduction" and "Materials and Methods" sections are developed and presented thoroughly and comprehensively. The results obtained are correctly described and clearly presented. The "Discussion" section greatly expands the analysis of their own results with an indication of their applicability, as well as an indication of limitations. The results obtained entitle the authors to formulate conclusions at the end of the "Summary" section and at the end of the "Discussion" section.
Here are some comments on the manuscript:
- I propose a minor correction to the title of the manuscript: “Investigation of genetic variants associated with tryptophan metabolite levels to serotonin and kinurenin pathways in patients with bipolar disorder”
- the use of the abbreviation "HPA" in line 67 is unnecessary, as this abbreviation is no longer used. Please provide the full name instead of the abbreviation
- in Table 1, in the row for "Sex (% F)," in the column for "HC (n = 45)" please change "64.4%" to "64.4"
- in the "Materials and Methods" section, I will ask for the date of the study and thus data collection
- in the "References" section for References 4, 15 and 21, please remove unnecessary capital letters in article titles.
Author Response
Q1) I propose a minor correction to the title of the manuscript: “Investigation of genetic variants associated with tryptophan metabolite levels to serotonin and kinurenine pathways in patients with bipolar disorder”
R1) We modified the title as suggested
Q2) the use of the abbreviation "HPA" in line 67 is unnecessary, as this abbreviation is no longer used. Please provide the full name instead of the abbreviation
R2) We made the suggested change
Q3) in Table 1, in the row for "Sex (% F)," in the column for "HC (n = 45)" please change "64.4%" to "64.4"
R3) We made the suggested change
Q4) in the "Materials and Methods" section, I will ask for the date of the study and thus data collection
R4) We added the requested information
Q5) in the "References" section for References 4, 15 and 21, please remove unnecessary capital letters in article titles.
R5) We made the suggested change
Reviewer 4 Report
The manuscript submitted by Pisanu et al. conducted a small-scale GWAS study that, up to my knowledge, is the first of its kind that specifically investigated genetic variants associated with tryptophan metabolite to the serotonin and kynurenine pathways in bipolar patients. Generally, the experiment is well-designed, and statistics are methodically conducted. Whilst the number of SNPs found and their significance values were relatively low, we should accept the fact that the small cohort size, as well as a potentially significant interference factor from the BMI, could all have effects on the output. This fact could be further demonstrated in both regression approaches, and the binary classification predictability was lower in the model that included genetic variants. Nevertheless, given the novelty of the experiment design and data presented, I consider the manuscript deserves its deed of publication for the interest of the greater community.
Author Response

(The authors gave the same response as above.)

Reviewer 5 Report
The article "Investigation of genetic variants associated with levels of metabolites of the tryptophan to the serotonin and kynurenine pathways in patients with bipolar disorder" is a very reliable, very well-written work with high substantive values ​​and good methodology. Therefore, I have no significant critical remarks and I congratulate the authors. The only remark I have about the introduction. I believe that it should be supplemented with information on the strong relationship between drug-resistant depression and bipolarism, on the one hand, and on the other hand, on the potential contribution of disorders of metabolism of the tryptophan to the serotonin and kynurenine pathways pathology in drug resistance in depression.
Author Response
Q1) The only remark I have about the introduction. I believe that it should be supplemented with information on the strong relationship between drug-resistant depression and bipolarism, on the one hand, and on the other hand, on the potential contribution of disorders of metabolism of the tryptophan to the serotonin and kynurenine pathways pathology in drug resistance in depression.
R1) We concur with this observation. The sentences: “Of note, there is consistent evidence that a poor response to antidepressants prescribed in acute depressive phases might be a predictor of later bipolarity (17850879; 29208408).” “Interestingly, the pathophysiological underpinnings of the treatment resistant forms of depression (often predicting a later BD) appear to include alterations in tryptophan and kynurenine pathway (28412922).”